# DISTILLED AGENT DQN
# FOR PROVABLE ADVERSARIAL ROBUSTNESS

## ABSTRACT

As deep neural networks have become the state of the art for solving complex reinforcement learning tasks, susceptibility to perceptual adversarial examples have become a concern. The transferability of adversarial examples is known to enable attacks capable of tricking the agent into bad states. In this work we demonstrate a simple poisoning attack able to keep deep RL from learning, and into fooling it when trained with defense methods commonly used for classification tasks. We then propose an algorithm called DadQN, based on deep Q-networks, which enables the use of stronger defenses, including defenses enabling the first ever on-line robustness certification of a deep RL agent.

## 1 INTRODUCTION

To ensure Reinforcement Learning (RL) behaves reliably in the real world, it is important to consider settings where an adversary aims to actively hinder the learning process of the agent. While prior work has explored robustness in the setting of discrete RL (Morimoto & Doya, 2005; Boyan & Moore, 1995), most recent progress in the field has focused on dealing with continuous states by using neural networks (Sutton et al., 2000; Mnih et al., 2013; Peters & Schaal, 2008).

In this work we present a new approach for training RL systems to be more (provably) robust. The key idea is to decouple the DQN network architecture into a policy (student) network S and a Q-network in a way which allows us to robustly train the policy network and use it for exploration while at the same time preserving the standard way in which the Q network is trained. We then show how to naturally incorporate state of the art defenses, developed in the context of deep supervised, learning to the reinforcement learning setting by training the student network in two ways: (i) via adversarial training with methods such as FGSM where we generate adversarial states that decrease the chance the optimal action is selected, and (ii) via provably robust training with symbolic methods which guarantee the network will select the right action in a given state despite any possible perturbation (within a range) of that state.

**Contributions**   Our main contributions are:

- The first deep RL algorithm, DadQN, designed to be adversarially defended with state-of-the-art adversarial training as well as provably robust training into our RL algorithm. To our best knowledge, this is the first time, on-line robustness certification has been achieved with a deep RL agent.
- An attack, illustrated in Figure 1, that hinders learning in modern deep RL algorithms.
- An evaluation demonstrating that DadQN can defend against adversarial attacks when defenses against DQNs catastrophically fail, while being of comparable performance when adversarial attacks are not present.

## 2 RELATED WORK

At roughly the same time as deep reinforcement learning methods were first being developed Schulman et al. (2015); Mnih et al. (2013) it was noticed by Goodfellow et al. (2014) that in the

---

[1] tag sale by PetalArt, money by DesignContest, crying face, microscope, shopping cart and graduation cap by Twitter, Inc., are licensed under CC BY 4.0.

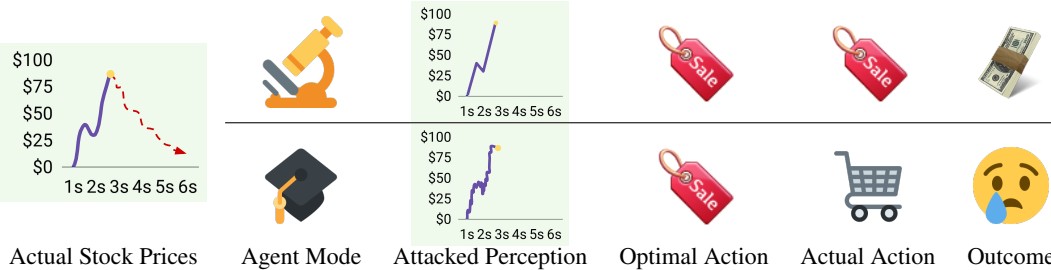

Figure 1: An illustration of the UQP Poisoning Attack on a hypothetical stock trading agent.[1]

setting of supervised learning, classifiers were susceptible to adversarial attacks by introducing imperceptible perturbations to the inputs. At the same time, it was determined that training using these adversarially crafted images could serve as a way to defend against them. Later, Carlini & Wagner (2016) determined that many defensive techniques were insufficient. A particularly successful defense is that of Madry et al. (2018). This method uses adversarial examples generated by the PGD attack as training examples (instead of the original examples). While it is experimentally effective, it provides no guarantees about the resulting network's robustness. Carlini & Wagner (2017) introduced further method for evaluating the robustness of neural networks. Katz et al. (2017) provided the first system which could (for very small networks) determine local-robustness, that is, at test time the network is proven to be either $\epsilon$-robust for an individual image. Gehr et al. (2018) used abstract interpretation to scale the local robustness analysis by sacrificing completeness. Raghunathan et al. (2018) introduced a technique to train networks to be certifiably robust (though again limited to small networks). Multiple recent techniques have been developed since to train increasingly larger networks to be certifiably robust including Wong & Kolter (2018); Mirman et al. (2018); Dvijotham et al. (2018); Wong et al. (2018). These techniques represent different points in the spectrum of classification accuracy, certifiable robustness, experimental robustness, and training speed.

While Pattanaik et al. (2017) used adversarial examples to train an agent and improve its experimental robustness, Behzadan & Munir (2017b) noticed that often doing this prevents learning. Behzadan & Munir (2017a) further demonstrated that this effect could be intentionally amplified to prevent learning intentionally on the part of an adversary. Gu et al. (2018) demonstrated a technique for training the A3C algorithm to be adversarially robust by training a parallel adversarial actor-critic pair for each protagonist actor-critic pair. The analysis of their technique is limited to noisy environment adversaries, and not intentional agent-aware adversaries such as FGSM. Similarly, Ferdowsi et al. (2018) created an adaptive adversary to improve robustness.

Pinto et al. (2017) first used adversarial training to improve robustness of deep reinforcement learning algorithms, specifically TRPO. Mandlekar et al. (2017) demonstrated techniques to construct physically plausible perturbations adversarially to improve deep RL. Gu et al. (2018) then demonstrated how adversarial training could also be used to improve A3C.

Papernot et al. (2015) first proposed that distilling a neural network by training the same architecture with the probability vector outputs of the original network might act as a defense against adversarial examples. While Carlini & Wagner (2016) found that defensive distillation is insufficient to protect against adversarial examples for supervised learning, our algorithm uses distillation in combination with more effective defenses to protect reinforcement learning. Chen et al. (2017) use a pre-defined rule-based teacher policy to assist a deep RL system. They introduce a new DQN algorithm which predicts when to consult the teacher and how to learn from the teacher's experiences. In our setting there is no known external teacher. Lin et al. (2017a) use distillation to improve reinforcement learning itself by introducing the collaborative asynchronous advantage actor-critic algorithm (cA3C). Their method allows for knowledge transfer between agents in cases where multiple agents are playing simultaneously in different environments with potentially different tasks. To do this, they use a "deep alignment network" which learns to transfer the outputs from a teacher network to a student network. Unlike our system, their system only improves agents learning from different tasks, and does not provide a method to defend the learned policy. Shashua & Mannor (2017) train a DQN, that avoids the worst case outcome and performs well in the presence of a non-deterministic environments by using an extended Kalman-Filter. However, their Kalman-update requires multiplication of matrices

of size quadratic in the total number of weights and thus does not scale to neural networks that play Atari-Games with the screen input. On the other hand, we optimize a policy to generalize to non-game effecting perturbations in a way which we show is scalable to very large neural networks. We furthermore demonstrate how to do this in a way that increases provability.

## 3 BACKGROUND

The goal of reinforcement learning is to determine a behavior policy[2] $\pi(a \mid s)$ for a given game $G\colon \text{State} \times \text{Action} \to \text{State} \times \mathbb{R}$, that maximizes the expected discounted sum of future rewards $Q^\pi(s) = \mathbb{E}[\sum_{t=1}^\infty \gamma^t r_t | s_1 = s, a_1 = a, \pi]$, where $\gamma \in [0, 1]$ is a discount factor and $s_{i+1}, r_i = G(s_i, a_i)$. The objective of Q-Learning (Watkins & Dayan, 1992) is to learn the function $Q^*(s)_a = \max_\pi Q^\pi(s)_a$ and construct a policy by greedily picking its optimal action.

### 3.1 DEEP Q LEARNING

In Deep Q-Learning (Mnih et al., 2015; 2013) the goal is to progressively approximate $Q^*$ by weights $\theta_i$ for a neural network $Q(s; \theta_i)_a$ at iteration $i$. Samples are generated and stored in an experience replay buffer $D_i = \{e_1, \ldots, e_i\}$ where $e_i = (s_i, a_i, r_i, s_{i+1})$ is generated using a sufficiently random agent based on the approximation $Q(s; \theta_i)$.

Integral to the DQN algorithm is the use of a network with lagging weights $\theta_i^- = \theta_{k\lfloor \frac{i}{k} \rfloor}$ for some $k$. The loss at iteration $i$ is described by

$$L_i = \mathbb{E}_{(s,a,r,s') \sim U(D_i)}[(r + \gamma \max_{a'} Q(s'; \theta_i^-)_{a'} - Q(s; \theta_i)_a)^2]$$

A few iterations of stochastic gradient descent are used to minimize $L_i$ and produce $\theta_{i+1}$, although often multiple exploration steps are used first.

### 3.2 ADVERSARIAL ATTACK

Unlike traditional reinforcement learning, the presence of adversarial attacks introduces the complication that the game has some level of knowledge about the agent playing the game and can act in specific ways knowing its behavior. The environment that our algorithm is designed to address is given by our definition of an *agent aware game*.

**Agent Aware Game**    In agent aware reinforcement learning, the game $G$ not only has access to the agent's choices, but also has access to the learning system and the agent itself. Thus, an adversarial agent's aim is to play the game in such a way as to poison learning, but also potentially play differently when it is aware if the RL system is training or testing.

We consider only *manipulative* agent aware games, defined to be games

$$G\colon \text{State} \times \text{Action} \times \text{Agent} \to \text{State} \times \mathbb{R}$$

which are composed of a core game $H\colon \text{State} \times \text{Action} \to \text{State} \times \mathbb{R}$ and a manipulative adversary $M\colon \text{State} \times \text{Agent} \to \text{State}$ such that $G(s_i, a_i, \pi) = (M(s_{i+1}^-, \pi), r_i)$ where $(s_{i+1}^-, r_i) = H(s_i, a_i)$, and such that for any $\pi$ where $s_{i+1} = M(s_{i+1}^-, \pi)$ it is true that $\forall a. H(s_{i+1}, a) = H(s_{i+1}^-, a)$. In other words, manipulation does not effect how the core game works or is played.

The attacks we consider are $L_\infty$ $\epsilon$-attacks (not to be confused with the $\epsilon$ used for greedy exploration). Specifically $M(s, \pi) \in \mathbb{B}_\epsilon(s) = \{s' \mid ||s - s'||_\infty \le \epsilon\}$, that is, $\mathbb{B}_\epsilon(s)$ is an $\epsilon$ sized $L_\infty$ ball.

We consider both, test time attacks, and attacks that change their behavior when the agent is training.

### 3.3 TESTING ATTACKS

Huang et al. (2017) attack a network policy trained by reinforcement learning, at testing time and find that traditional untargeted attacks such as FGSM are sufficient to significantly reduce the performance

---

[2]We write $\pi(s)$ to mean vector of scores for each action in the deterministic greedy policy where $\arg\max_a \pi(s)_a$ is the action taken. When written this way, it is assumed that $\pi$ is a neural network.

of the policy produced by DQN, A3C and TRPO. Lin et al. (2017b) used targeted attacks to "enchant" the agent into performing specific actions. We evaluate only with untargeted FGSM attacks.

## 3.4 TRAINING ATTACKS

Attacking the training procedure such that the model never learns is known as poisoning Yang et al. (2017); Steinhardt et al. (2017); Biggio et al. (2012). Behzadan & Munir (2017a) introduce an attack used to prevent the DQN from ever learning the correct $Q$ function. In their attack, they do not assume the attacker has direct access to $\theta_Q$ and instead train a version of it, $\theta_{Q'}$, in parallel. They learn an "adversarial policy" which picks an action $a'$ far from the optimal one and then they perturb the observed next-state $s'$ to cause $Q(s'; \theta_{Q'}^-)$ to be maximized (they use the non Double DQN variant) for the action $a'$ in hopes the attack transfers to the intended DQN.

We introduce an attack, Untargeted Q-Poisoning (UQP), which does not need to train additional networks, and which has access to the agent's network. We allow the attack the to switch its behavior when the agent is being tested in order to simulate the inevitable asymmetry between production and development environments and common subsequent dysfunction it is known to cause. In UQP during training time, an attack state is chosen in the style of FGSM to reinforce the decision of the agent policy, thus often creating an illusion of highly successful training. During test time, standard FGSM is used where the "classification" is chosen by the policy. We define it as a manipulative adversary:

$$M(s,\pi) = \begin{cases} s - \alpha\mathrm{sign}(\nabla_s\mathcal{H}(g, \pi(s))) & \pi(s) \text{ is learning and } g = \arg\max_a \pi(s)_a \\ s + \alpha\mathrm{sign}(\nabla_s\mathcal{H}(g, \pi(s))) & \text{otherwise and } g = \arg\max_a \pi(s)_a \end{cases}$$

where $\mathcal{H}$ is the cross entropy between the probability distribution $\pi(s)$, and the optimal action (encoded as one hot vector) from that probability distribution. This attack is visualized in Figure 1. Here the adversary is aware of whether the agent is training or being actively used, and is able to change the agent's perception of the stock histories. When it is training, it might show the agent stock values with less noise than normal. When the agent is then being actively used to trade, it would use the standard FGSM attack to convince it to buy when it should sell. Intuitively, by using the negative of the gradient during training, the UQP attack increases the over-estimation of the value of an action, which Van Hasselt et al. (2016) has determined to decrease learning performance. The negative of the gradient has the added benefit of counteracting adversarial training. While UQP attack is aware of whether the agent is testing or training, our results show that it is only necessary to use during training to derail the performance of a DQN, and thus is functionally a poisoning attack.

## 4 DISTILLED AGENT DQN (DADQN)

In contrast to deep supervised learning, deep reinforcement learning is more resource intensive task, and has less stable training dynamics, even for seemingly simple problems. As such, it is important that any proposed defense mechanism is efficient and usable in an online setting and induce as little perturbation as possible to standard network training algorithm.

Furthermore, while some progress has been made towards making deep RL experimentally robust, so far there has been less work on certifiable robustness. Towards this, we propose a method to leverage the base Q learning algorithm's ability to learn the correct Q function given a sufficiently random exploration agent (Even-Dar & Mansour, 2002; Bertsekas, 2008; Tsitsiklis, 1994; Watkins & Dayan, 1992).

Rusu et al. (2015) first described the method of improving the learned policy in Deep-Q Learning using *Policy Distillation* (PD). In this technique a $Q$-approximation is first learned by the standard DQN algorithm. New games are then played using $Q$ as a greedy policy, and the states $s$ are recorded. A student network $S$ is then trained on these states to mimic the behavior of $Q$. Much like PD, our method involves distilling a policy network from $Q$. Standard PD only uses the distilled policy during test time, and has so far only been tested in this way. Here we introduce DadQN which trains the student policy $S$ at the same time as it is learning the $Q$ network and uses the student policy instead for exploration. By decoupling the policy network from the $Q$ network, we are able to train the policy with additional defensive constraints and use it for the exploration agent in addition to testing with it. Given a loss $L_D$ and learning rates $\alpha_Q, \alpha_S \in (0, 1)$ our algorithm is as follows:

---

**Algorithm 1** DadQN pseudocode

---

Initialize a state $s$, weights $\theta_Q, \theta_S$
**for** $i = 0, \ldots$ **do**
   **for** $j = 0, \ldots, n$ **do**
      Pick an an action $a$ with a fair strategy (ex. $\epsilon$-greedy) with $\pi_{\theta_S}$ based on $S(s; \theta_S)$
      Play the game $(s', r) := G(s, a, \theta_S)$
      Store $(s, a, r, s')$ in $D$
      $s := s'$
      **if** $i \mod k == 0$ **then**
         $\theta_Q^- := \theta_Q$
      **end if**
      Pick a batch $\mathcal{D} \sim U(D)$
      Train the underlying Q:
         $\theta_Q := \theta_Q - \alpha_Q \nabla_{\theta_Q} \sum_{(s,a,r,s') \in \mathcal{D}} (r + \gamma Y - Q(s; \theta_Q)_a)^2$
      Train the student $S$ from $Q$:
         $\theta_S := \theta_S - \alpha_S \nabla_{\theta_S} \sum_{(s,a,r,s') \in \mathcal{D}} \left[ L_D(s, \theta_Q, \theta_S) \right]$
   **end for**
**end for**

---

Here $Y \equiv Q(s'; \theta_Q^-)_{\arg\max'_a Q(s';\theta_Q)'_a}$ is the Double DQN (Van Hasselt et al., 2016) next $Q$ estimate.

The intuition behind this algorithm is that the loss for the Q function has not been changed and thus after being trained on sufficiently many random paths, the $Q$ function will approach $Q^*$ regardless of what the student learns. Assuming the student will learn no matter what it is initialized with, the student should be able to handle the concept-shift of $Q$ learning and incorporate the provided constraint. Presumably, the additional constraints allow the student to achieve a better score when playing, so the student should explore higher reward paths.

While $L_D$ could potentially be any optimizable function, we evaluate with either one of the defensive loss described in Section 4.1, or an undefended mean squared error (MSE) loss $L_{\text{MSE}}$. While Rusu et al. (2015) observed that KL divergence works better than MSE and negative-log-likelihood experimentally, we found MSE to be sufficient, simple and common enough for our evaluation.

Wang et al. (2015) observed better learning performance by introducing a specific Q network architecture, the Dueling DQN, which is split into two components: an advantage network $A(s; \theta_Q) : \mathbb{R}^m$ computing the relative advantage of the $m$ actions, and a value network $V(s; \theta_Q) : \mathbb{R}$. These are combined as $Q(s; \theta_Q) = V(s; \theta_Q) + (A(s; \theta_Q) - \frac{1}{|\mathcal{A}|} \sum_{a \in \mathcal{A}} A(s; \theta_Q)_a)$, where $\mathcal{A}$ denotes the action space. The advantage and value are defined to share early network layers. We note that the output of $A$ alone is sufficient to replicate a greedy policy based on $Q$. However, we observe that the value network may be able to discriminate between states earlier on in the learning than the advantage network. We thus train both the advantage network and value network of the student from the $Q$ independently with the same loss, defining $L_{\text{MSE, Dueling}}$ for use as $L_D$ in Algorithm 1 as follows:

$$L_{\text{MSE, Dueling}}(s, \theta_Q, \theta_S) = ||A_Q(s; \theta_Q) - A_S(s; \theta_S)||_2^2 + ||V_Q(s; \theta_Q) - V_S(s; \theta_S)||_2^2. \quad (1)$$

**The Exploration Agent** Originally, randomization in exploration was accomplished for the DQN using $\epsilon$-greedy search (Watkins, 1989). Here, the action $\arg\max_a S(s; \theta_S)_a$ is used with probability $1 - p$ and an action is picked uniformly with probability $p$ at every time step (we use $p$ instead of $\epsilon$ so to not overload $\epsilon$ used in attacks). If $p \geq c$ for a constant $c > 0$, then standard $Q$ learning is known to converge. $p$ is decreased linearly with the number of frames played until it reaches a fixed $c$.

While this method has good theoretical guarantees, Fortunato et al. (2017) noticed improvements by letting the Q-network learn a mean and variance on noise for the weights in its dense layers. We experiment with the same idea here, however only the student network $S$ learns with noise. We notice that because the student network lags behind the $Q$ network, it might require more diverse samples even when $Q$ has sufficiently learned the correct behavior. We thus introduce an exploration noise constant $\eta \geq 1$, which we multiply with the learned weight variances at exploration time.

### 4.1 Defending DadQN

The primary motivation for decoupling the DQN network into a policy-student network $S$ and a $Q$-network is to allow one to leverage additional constraints on the student network without affecting the learning of the correct $Q$ function (provided that exploration is sufficiently noisy). We address the problem of adversarial robustness in reinforcement learning by improving $S$'s robustness. Specifically, we split $L_D$ into a defense $\alpha$ and a defense loss $L_O$. The defense $\alpha$ uses $\theta_S$ and the sample $s$ to produce a *defense goal* $s_d$ (which could be a concrete or symbolic) which is then passed to $L_O$.

$$L_D(s, \theta_Q, \theta_S) = L_O(s, s_d, Q(s, \theta_Q), \theta_S).$$

**Adversarial Training**    A variety of techniques have been developed for increasing the robustness of neural networks, typically by training with adversarial examples (Tramèr et al., 2017; Shaham et al., 2015; Madry et al., 2018). Rather than providing the $Q$ network with adversarial examples as in Mandlekar et al. (2017), we provide them (in this case, as concrete defense goals) $s_d$ to the student:

$$\alpha_{\epsilon,\mathrm{FGSM}}(\theta_S, s) = \mathrm{FGSM}_\epsilon(s, \arg\max_a S(s; \theta_S)_a, \theta_S)$$

Here, $\mathrm{FGSM}_\epsilon$ produces a concrete adversarial example $s_d$ in the $\epsilon$ sized $L_\infty$ ball around $s$ that makes the best action in $s$ as least likely to be selected in $s_d$ as possible (that is, it is an untargeted attack).

Finally, we use the traditional MSE loss function:

$$L_{O,\mathrm{MSE}}(s, s_d, \bar{q}, \theta_S) = ||\bar{q} - S(s_d; \theta_S)||_2^2.$$

In practice, we use this loss with probability $p$, the rest of the time using the non-attacked point $s$ instead of $s_d$. For Dueling DQNs we calculate the advantage and value separately as in Equation 1.

**Provable Robustness Training**    In addition to adversarial training, we train and certify the robustness of our networks with the DiffAI framework introduced by Mirman et al. (2018) using its Interval domain. This framework has been shown capable of training networks on the scale of DQNs used in Atari with minimal speed and memory overheads over undefended SGD. Additionally, it has been shown to decrease the accuracy of networks very little, which is essential to reinforcement learning where unstable dynamics can be an issue. Formally, we first create the abstract (symbolic) defense goal $s_d = \mathbb{B}_\epsilon(s)$. We then use DiffAI to soundly propagate the defense goal $s_d$ through the network $S$ (via symbolic computation), obtaining a symbolic element $g_f$ as a result. Finally, DiffAI defines a differentiable loss $L_I : \mathrm{Interval} \times \mathbb{N} \to \mathbb{R}$ which takes as input the final element $g_f$ and a target (action in our case). The loss has the property that for some target $t$ if $L_I(g_f, t) \leq 1$ then $\forall s' \in \mathbb{B}_\epsilon(s).\ \arg\max_a S(s'; \theta_S)_a = t$. That is, in this case, we have proved that any element inside the $\epsilon$ sized $L_\infty$ ball around $s$ will be classified to the action $t$ by the student network. We define our defensive loss as a combination of this loss and the adversarial loss described earlier for a concrete $\alpha$:

$$L_{O,\mathrm{Interval}}(s, s_d, \bar{q}, \theta_S) = L_{O,\mathrm{MSE}}(s, \alpha(\theta_S, s), \theta_S) + \lambda L_I(g_f, \arg\max_a \bar{q}_a)$$

where $\lambda \geq 0$ is the constant with which we want to prioritize the DiffAI loss.

## 5 Experimental Evaluation

We now present our experimental results comparing DadQN to existing methods. We note that while it may be beneficial to *intentionally* train one's own DQN by simulating an adversarially attacking manipulator (in an attempt to defend it), there are learning time attacks which prevent this from being effective. Thus, central to our analysis is the notion that the agent does not know whether it will be attacked during training and with what attack, or whether it will be attacked during testing and with what attack. We show that in each situation, DadQN is capable of providing a stronger or equally strong a defense as existing work, and virtually always avoids the worst case failures. We also show the first improvement in provable robustness in DQNs by using DiffAI as a training defense.

### 5.1 Experimental Setup

We tested with 3 Atari games (Bellemare et al., 2013) from the OpenAI Gym (Brockman et al., 2016): RoadRunner (RR), Pong and Boxing. Every 10 episodes we play a validation game. In a these games

Table 1: Validation game reward comparison for UQP poisoning attack.

| Game | Test Attack | DQN | Untargeted Quality Poisoning Training Attack | | | |
|---|---|---|---|---|---|---|
| | | | DQN | DadQN | DadQN + FGSM Def | DadQN + DiffAI |
| RR | none | 46821.66 | 17940.54 | 25227.04 | 30513.18 | 18903.09 |
| | FGSM$^{p=0.4}$ | 15517.12 | 15487.56 | 26516.17 | 30766.76 | 20954.75 |
| Pong | none | 20.62 | 12.23 | 19.19 | 18.99 | 11.14 |
| | FGSM$^{p=0.4}$ | -14.43 | 18.61 | 18.10 | 19.10 | 10.55 |
| Boxing | none | 87.07 | 41.07 | 82.15 | 93.30 | 46.82 |
| | FGSM$^{p=0.4}$ | 67.11 | 40.45 | 78.14 | 93.52 | 51.52 |
| Breakout | none | 263.66 | 26.69 | 30.77 | 130.98 | - |
| | FGSM$^{p=0.4}$ | 7.91 | 14.53 | 54.22 | 125.29 | - |

Table 2: Comparing defended DQN to defended DadQN by validation game score.

| Game | Test Attack | DQN trained with... | | | | DadQN trained with... | | | |
|---|---|---|---|---|---|---|---|---|---|
| | | none | TrAtk | Def | TrAtk+Def | nothing | TrAtk | Def | TrAtk+Def |
| RR | none | 20244 | 15903 | 242 | 820 | 13315 | 22726 | 19781 | 18480 |
| | FGSM$^{p=1}$ | 780 | 19478 | 981 | 1101 | 647 | 15234 | 18538 | 17566 |
| Pong | none | 19.85 | 17.83 | -21.00 | -20.86 | 20.55 | 19.73 | 20.31 | 18.29 |
| | FGSM$^{p=1}$ | -21.00 | 17.42 | -21.00 | -20.94 | -19.04 | 13.50 | 19.12 | 17.03 |
| Boxing | none | 77.71 | 41.12 | -26.67 | -23.40 | 95.49 | 80.60 | 79.88 | 73.11 |
| | FGSM$^{p=1}$ | 8.60 | 41.05 | -9.20 | -52.77 | 5.07 | 55.29 | 56.30 | 66.75 |

we verify the decision made by the network using the Box domain. We binary-search for the largest $\epsilon$ such that the decision is safe. Validation games alternate between never attacking, or attacking with some probability every timestep. Each experiment was run for 4 million frames.

To compare DadQN with DQNs, we implemented a variety of extensions known to increase training performance when enabled in concert as in RainbowDQN (Hessel et al., 2017): Priority Replay (Schaul et al., 2015), DoubleDQN (Van Hasselt et al., 2016), DuelingDQN (Wang et al., 2015), and NoisyNet (Fortunato et al., 2017) using noise constant $\eta = 2$ as described in Section 4. During validation episodes we disable noise due to NoisyNet and use $\epsilon$-greedy exploration with $\epsilon = 0.005$. Due to limited resources, we choose parameters for the three games that yielded faster training than those reported by RainbowDQN [3]. All parameters can be found in Tables S1 and S2 in the Appendix.

We implemented both DadQN and DQN in PyTorch (Paszke et al., 2017) asynchronously (Mnih et al., 2016). Depending on the parameters and the hardware one run took between 4 to 30 hours, and the majority took about 20 hours. We evaluated using Nvidia 1080Tis and Nvidia k80s, with otherwise modern hardware. On our fastest machine, using a Nvidia 1080Ti, our implementation of DQN plays 266.9 frames per second and the DadQN algorithm 266.0 frames.

**Attacks & Defenses** We consider both untargeted FGSM, as well as our UQP attack. For all uses of FGSM we use $\epsilon = 0.004$. During validation games we either do not use any attacks whatsoever as in rows preceeded by "none", or we use FGSM with probability $0.4$ or $1.0$, written as FGSM$^{p=0.4}$ or FGSM$^{p=1}$. When the UQP attack is used during validation games, it is functionally equivalent to the FGSM attack, and thus for simplicity we only write FGSM for testing attacks. For standard Q-Networks the semantics of attacking and defending with FGSM are similar but not quite the same: when attacked (*TrAtk*), the environment produces adversarial examples which are then stored in the replay buffer and are seen many times in training. When used as a defense (*Def*), the perturbation is applied to a training example when it is sampled from the replay buffer, thus each perturbation will only be seen once. We apply our attacks and defenses to the 4-stacks of consecutive frames used for training DQNs on Atari. While single frame attacks conceptually fit the setting, we are primarily evaluating the effectivity of defenses and thus choose pick the most efficient attck – which is attacking 4-stacks (as used in Behzadan & Munir (2017b)).

## 5.2 POISONING AS AN ATTACK

Table 1 shows the average final episode score (weighted by number of frames) from the 15 consecutive best (by sum score) validation episodes during training. See Appedix B for the details of this weighting.

---

[3]Hessel et al. (2017) report parameters which are a compromise between many games.

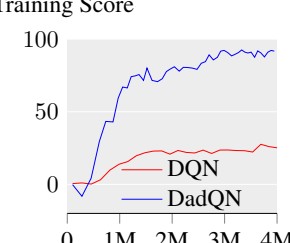

Figure 2: UQP poisoning attacks with number of frames played on x axis.

Table 3: Avg. max provable $\epsilon$ with DiffAI Box.[4]

| Game | DQN trained with... | | DadQN trained with... | |
|---|---|---|---|---|
| | none | TrAtk | Def | DiffAI |
| RR | 2.66e-08 | 1.76e-06 | 8.42e-05 | 7.82e-04 |
| Pong | 3.71e-08 | 1.98e-06 | 2.19e-05 | 4.11e-04 |
| Boxing | 2.86e-07 | 3.67e-07 | 9.20e-07 | 5.12e-05 |

The leftmost DQN column first shows these scores without any attack on DQNs for reference. When attacking DQN during training we observe significant drop in score for unattacked validation games. We can see from this table that UQP is a strong attack against DQN's as the presence of the attack (i) impacts undefended training performance and (ii) significantly impacts the performance of the agent in unattacked *and* attacked games during testing. To exemplify (i), Figure 2 shows the training reward for Boxing in the presence of an attack.

## 5.3 UTILITY OF DEFENDING WITH DADQN

Table 2 shows the average final episode score (weighted by number of frames) using the network weights from the best validation game during training over 15 validation episodes. We can see that attacking a DQN which has never encountered adversarial examples before (*none* column under DQN) drastically lowers its score. When attacked by the environment (*TrAtk*), whether intentionally or not, the agent learns to play the game, although not as well as without an attack or the DadQN agent that has been defended, or DadQN being defended *and* attacked. We also see that a DQN using the previously defined FGSM training defense (*Def*) - perturbations happen after sampling from the replay buffer) does not learn how to play the game whatsoever. We hypothesize that the attack variant places the adversarial examples in the replay buffer making it easier for the agent to learn since it then encounters the same perturbations more frequently. DadQN trained with the FGSM defense performs nearly as well as baseline DQN without attacks, and much better than baseline with attacks.

In addition to increasing test time attack robustness, we can see in Table 1 that DadQN trained without an explicit defense is robust to poisoning. Rewards for DadQN were nearly always greater than DQN whether playing finally in an attacked or unattacked game. Using the FGSM Defense amplified the effect to recover nearly the full performance of an unattacked DQN.

## 5.4 TOWARDS PRACTICAL PROVABLE ROBUSTNESS OF DQNS WITH DADQN

For DiffAI we explored a slightly different domain for each game. For Roadrunner we use $\lambda = 0.001$ with a non-defensive $\alpha(s, \theta_S) = s$ and $L_{O,\text{MSE}}$. Pong and Boxing instead use $\lambda = 0.01$ and $\epsilon = 0.0001$ with $\alpha_{0.004}$. For boxing, we use an attack probability of $0.4$. Knowing that DiffAI can decrease classification accuracy, some loss in score was expected. However, the last column of Table 1 demonstrates that DiffAI used in this was still a powerful defense against test the test time FGSM attack the UQP attack compared to otherwise undefended DQNs.

Table 3 demonstrates the average maximum $\epsilon$ which could be proven robust using DiffAI over the best (by aggregate score) 15 consecutive validation games. From this table, we can see that the best $\epsilon$ that DiffAI was able to prove without using any provability training was a tenth the size of the largest which could be found in DiffAI defended networks. Without any defense, the baseline DQN is not robust, and thus has a provability $\epsilon$ of less than a ten-thousandth the minimally normally expressible perturbation in 8-bit images (this would be $\epsilon \sim 0.004$). DiffAI on the other hand comes significantly closer than any system we are aware of, at nearly $25\%$ this value.

---

[4]Found with binary search between 0 and 1 with a maximum of 20 iterations or maximum accuracy of $1e - 6$ using a running average starting point.

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

## A HYPER-PARAMETERS

Table S1: Hyper-parameters used in the experiments. $\rightarrow$ indicates linear annealing.

| | RoadRunner | Pong | Boxing/Breakout |
|---|---|---|---|
| Optimizer | | Adam | |
| Adam-$\epsilon$ | | 0.00015 | |
| Learning rate | | 6.25E-05 | |
| Batch-size | | 32 | |
| Clip reward to sign | | True | |
| Frame-Stack | | 4 | |
| $\gamma$ | | 0.99 | |
| Priority Replay $\alpha$ | | 0.5 | |
| Priority Replay $\beta$ | | $0.4 \rightarrow 1.0$ over 100000 frames | |
| Target net Sync | | every 2000 frames | |
| Q-Net L2-Weight regularization | 0.0001 | 0 | 0 |
| NoisyNet Explore Constraint $\eta$ | 1 | 4 | 4 |
| Frames before learning | 10000 | 80000 | 80000 |
| Size of replay buffer | 100000 | 120000 | 200000 |
| $\epsilon$-greedy | 0 | $1.0 \rightarrow 0.0$ over 20000 frames | $1.0 \rightarrow 0.0$ over 20000 frames |

Table S2: Hyper-parameters for the student algorithm used in the experiments.

| | RoadRunner | Pong | Boxing | Breakout |
|---|---|---|---|---|
| Student Learning Rate | 0.001 | 2.00E-05 | 2.00E-05 | 6.25E-05 |
| Discard $V$ | True | False | False | False |
| Q-Network is NoisyNet | | False | | |
| Student-Net L2-Weight regularization | 0.001 | 0 | 0 | 0 |

## B WEIGHING OF SCORES & ADDITIONAL EXPERIMENTAL DATA

Some games, such as roadrunner, have multiple levels which is a non trivial problem for DQNs. As such, when the agents achieved a certain score, it then proceeded to play episodes it wasnt as adept at. These episodes achieved a low score and thus decrease the aggregates (such as mean or sum) over multiple episodes. It would be unfair to compare directly against an agent which had in fact never reached the second level. Thus we introduce a weighing scheme to combat this.

The values in Table 1 are calculated by

$$\max_{e \in \{1,...,E-15\}} \frac{\sum_{i=e}^{e+15} \texttt{episode\_score}[i] \cdot \texttt{episode\_frames}[i] \cdot A[i]}{\sum_{i=e}^{e+15} \texttt{episode\_frames}[i] \cdot A[i]}$$

where $E$ is the number of validation episodes played, $\texttt{episode\_score}[i]$ is the score earned in validation episode $i$, $\texttt{episode\_frames}[i]$ is the number of frames played in validation episode $i$ and $A[i]$ is an indicator whether the validation game was played with or without attacks. Table 2 uses the same weighing but over 15 new games, played from the best set of network weights.

To illustrate the importance of this weighing and show that it does not impact the qualitative interpretation of the results we provide unweighted scores for Table 1 in Tables S3 and S4. Table S3 shows

$$\frac{\sum_{i=e'}^{e'+15} \texttt{episode\_score}[i] \cdot A[i]}{\sum_{i=e'}^{e'+15} A[i]}$$

where $e'$ is the validation episode that maximized Equation B for the corresponding value. Table S4 on the other hand shows

Table S3: Validation game reward comparison for UQP poisoning attack, unweighted over same episodes as Table 1.

| Game | Test Attack | DQN | Untargeted Quality Poisoning Training Attack | | | |
|------|-------------|-----|------|-------|----------------|----------------|
| | | | DQN | DadQN | DadQN + FGSM Def | DadQN + DiffAI |
| RR | none | 12242.86 | 8085.71 | 5971.43 | 15185.71 | 9100.00 |
| | FGSM$^{p=0.4}$ | 5300.00 | 6900.00 | 9000.00 | 10257.14 | 7371.43 |
| Pong | none | 20.63 | 13.86 | 19.29 | 19.13 | 11.43 |
| | FGSM$^{p=0.4}$ | -14.57 | 18.71 | 18.29 | 19.14 | 11.25 |
| Boxing | none | 87.43 | 43.75 | 84.14 | 93.29 | 48.71 |
| | FGSM$^{p=0.4}$ | 67.57 | 42.25 | 78.57 | 94.13 | 54.43 |
| Breakout | none | 187.29 | 12.43 | 16.14 | 46.71 | - |
| | FGSM$^{p=0.4}$ | 3.43 | 9.29 | 23.57 | 41.00 | - |

Table S4: Validation game reward comparison for UQP poisoning attack, unweighted maximum.

| Game | Test Attack | DQN | Untargeted Quality Poisoning Training Attack | | | |
|------|-------------|-----|------|-------|----------------|----------------|
| | | | DQN | DadQN | DadQN + FGSM Def | DadQN + DiffAI |
| RR | none | 18900.00 | 8414.29 | 12085.71 | 15185.71 | 10642.86 |
| | FGSM$^{p=0.4}$ | 6671.43 | 10000.00 | 12612.50 | 15257.14 | 12328.57 |
| Pong | none | 20.63 | 13.86 | 19.29 | 19.13 | 11.43 |
| | FGSM$^{p=0.4}$ | -14.57 | 18.71 | 18.29 | 19.14 | 11.25 |
| Boxing | none | 87.43 | 43.75 | 84.14 | 93.29 | 48.71 |
| | FGSM$^{p=0.4}$ | 68.00 | 42.25 | 80.14 | 94.13 | 54.43 |
| Breakout | none | 214.00 | 15.14 | 18.86 | 53.57 | - |
| | FGSM$^{p=0.4}$ | 4.71 | 10.86 | 26.75 | 54.57 | - |

$$\max_{e \in \{1,\dots,E-15\}} \frac{\sum_{i=e}^{e+15} \texttt{episode\_score}[i] \cdot A[i]}{\sum_{i=e}^{e+15} A[i]}.$$

For Table S4 observe the same trends as for Table 1. However in Table S3 we see that the scores for RoadRunner especially are a lot lower for DadQN. The reason for this is that the agent reaches the second level, in which it does poorly; thus decreasing the score.

## C    UQP AGAINST ACTOR-CRITIC

Here we demonstrate that UQP is an affective attack against two actor-critic reinforcement learning algorithms for neural networks, A3C introduced by Mnih et al. (2016) (implemented synchronously) and PPO introduced by Schulman et al. (2017). We implement our attack on top of the framework provided by Kostrikov (2018), using the learning parameters as suggested for each algorithm. In order to demonstrate the weakness of these algorithms to this attack, we only test with the attack applied to the last image in the stack rather than all four images in the stack. In these attacks, unless otherwise specified we use an attack $\epsilon$ of 3 out of the possible range of $[0, 255]$.

For training, we either use no attack (shown in red), or the UQP attack with 4 iterations with step size 2 projecting back into the $L_\infty$ ball after every iteration as in PGD (shown in blue the figures). Figures S1, S3, S5 and S3 show the training scores sampled frequently, with the number of training steps in the x-axis and the score on the y-axis. The solid lines correspond to the average score from the last 10 episodes (unweighted), whereas the same-colored regions surrounding them show the max and min score of a game from the last 10 episodes.

For testing, we either use no attack, or the PGD attack with 16 iterations with step size 2 projecting back into the $L_\infty$ ball after every iteration as in PGD. In Figures S2, S4, S6, and S4 show the testing scores sampled significantly less frequently, at the number of training steps in the x-axis and the score on the y-axis. Each point correspond to the average score from 10 testing episodes (unweighted) at that time-step.

From these results it can clearly be seen that UQP is able to both prevent both actor-critic methods from learning to play the training set, and is able to keep the agent from being as robust when it is tested against a training set.

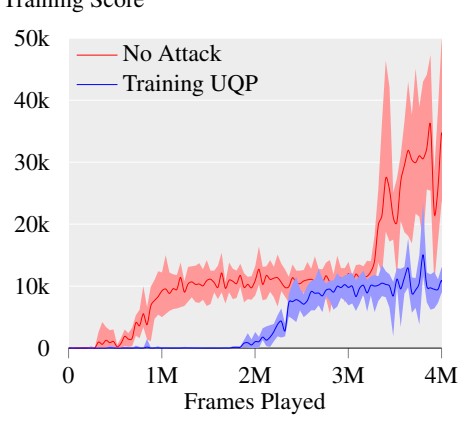

Figure S1: RoadRunner A3C Training

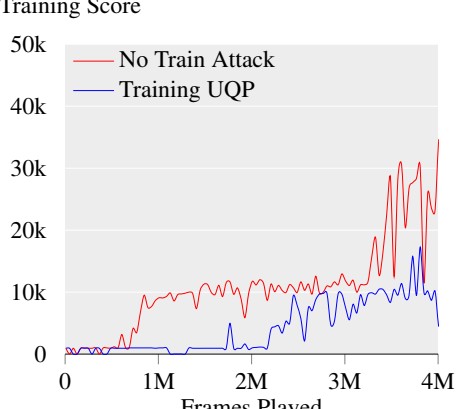

Figure S2: RoadRunner A3C Training

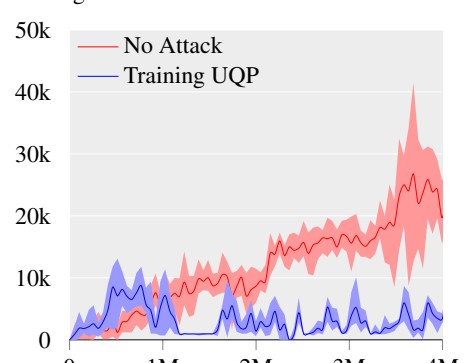

Figure S3: RoadRunner PPO Training

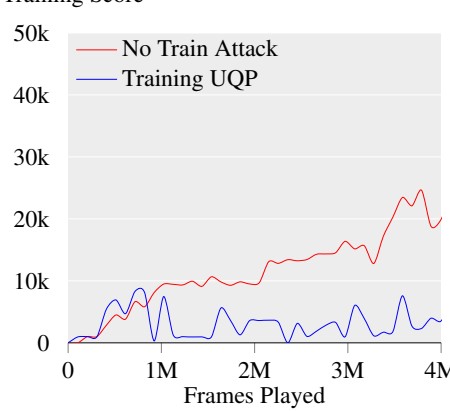

Figure S4: RoadRunner PPO Testing

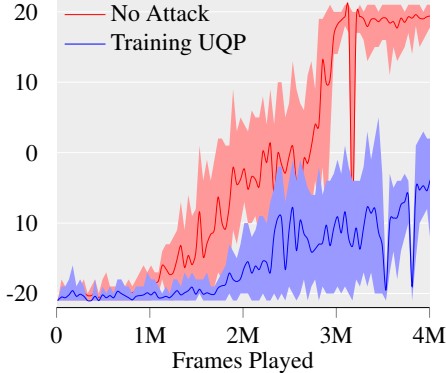

Figure S5: Pong A3C Training ($\epsilon = 9/255$)

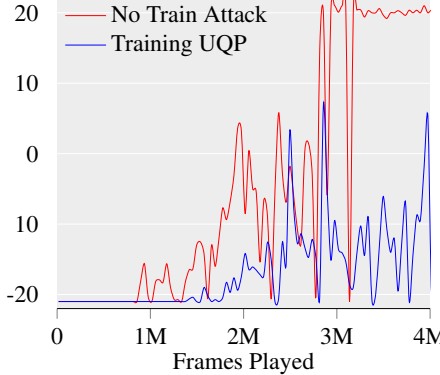

Figure S6: Pong A3C Testing ($\epsilon = 9/255$)

Training Score

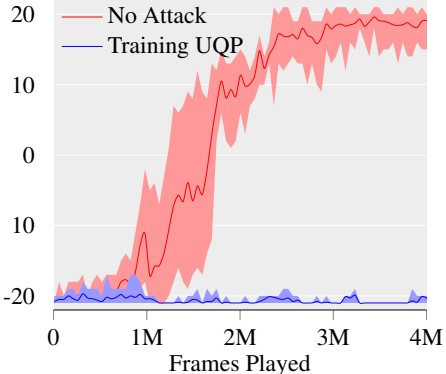

Figure S7: Pong PPO Training

Training Score

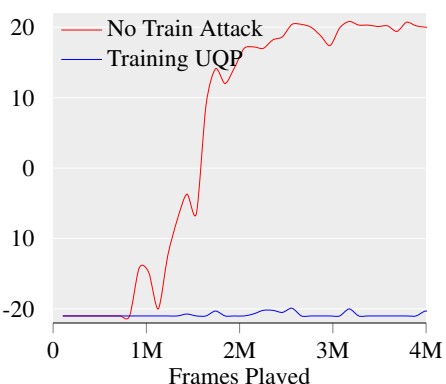

Figure S8: Pong PPO Testing

