# OpenReview forum: "Distilled Agent DQN for Provable Adversarial Robustness"
_ICLR.cc/2019/Conference_

### Official Review · AnonReviewer2 · 2018-11-02
**This paper contains interesting experiments, but it seem that the proposed methods lacks understanding**

**Rating:** 4
**Confidence:** 2

**Review:**

This paper considers adversarial attack and its defense to DQN. Specifically, the authors propose a poisoning attack that is able to fool DQN, and also propose a modification of DQN that enables the use of strong defense. Experimental results are provided to justify the proposed approach.

Detailed comments:

1.  Although the attack approach seems easy to implement, it would be interesting to see why it works. It might make this paper better if the intuition of the UQP is provided. FGSM is a well-known attack for deep learning models. What is the intuition of using the sign of the gradient of the cross-entropy? Since the argmax is a one-hot vector, this cross-entropy seems ill-defined. How to compute the gradient?

2. It would also be interesting to see why taking actions based on the student network enables better defense.  In DADQN, the authors seem to combine a few tricks proposed by existing works together. It might be better to highlight the contribution and novelty of this approach.

---

> ### Author Response · Authors · 2018-11-26
> **Responses**
>
> → Although the attack approach seems easy to implement, it would be interesting to see why it works. It might make this paper better if the intuition of the UQP is provided.
>
> Figure 1 in the paper highlights some intuition about why the UQP works.  In the top row, one can see that the UQP would cause the agent to see a less noisy than usual stock market, reinforcing its preexisting beliefs and leading it to not be robust when it eventually is shown a testing adversarial example as in the bottom row.  In section 2.4 we discuss the intuition: UQP reinforces the agent’s decisions (by perturbing by the negative gradient sign rather than the positive gradient sign).
>
> → What is the intuition of using the sign of the gradient of the cross-entropy?
>
> In two parts, we will first answer why we use the negative gradient of the cross entropy, and then why we use the sign of the gradient.
>
> (i)  The negative gradient of the cross entropy between the proposed distribution, $\pi(s)$ and the expected distribution $\argmax_a \pi(s)_a$ (implicitly encoded as a one hot vector as is common), is used as step of gradient descent in the direction which minimizes this cross entropy.  By minimizing this loss, an image is produced which will make the network signal even more confidence in the action that it would have already taken.  One effect of premature confidence is that it theoretically decreases the amount of exploration which occurs, even with epsilon-greedy training.  Further, in Q networks, training labels are essentially generated by the network itself.  Introducing over estimations in later time-steps can propagate error to estimates for earlier time steps. See Van Hasselt et al. (2016) for a discussion on one source of overestimation in Q networks and its effects.
>
> (ii)  The sign of the gradient is used instead of the gradient for normalization with respect to the L_infinity metric. In Explaining and Harnessing Adversarial Examples by Goodfellow et al. (2014) where FGSM is introduced, it is observed that a network might attempt to defend against gradient based attacks by inducing extremely small gradients in itself.  By using the sign of the gradient, one ensures that all gradients lead to perturbations of usable magnitude.  When attacking with respect to a ball of an L_p norm of width epsilon, it is intuitive that the best attack is the largest perturbation allowable in that ball.  By normalizing the gradient with respect to L_p and multiplying it by epsilon (or alpha in the case of our paper), one ensures the maximum size perturbation. The sign function is equivalent to the normalization function for the L_infinity norm.
>
> → Since the argmax is a one-hot vector, this cross-entropy seems ill-defined.
>
> Cross entropy is typically formally defined as taking two probability distributions.  A one-hot vector can be interpreted as a probability distribution.  As is common in the literature and in neural network libraries, one can either interpret the cross entropy as being between a probability distribution and a label intended to have probability 1.  One possible confusion here might arise from us having written the arguments to the cross-entropy swapped from what the order they usually appear.  This has been fixed in the revision.
>
> → How do you compute the gradient?
>
> The derivative of argmax (defined as the maximum coordinate of a finite vector) is defined on all but a measure zero set.  Automatic differentiation libraries such as pytorch return a value of zero for the “derivative” in the cases where it is not technically defined, although these cases are extremely rare.  In the case of UQP, one can consider $\argmax_a \pi(s)_a$ to be a fixed constant (not depending on $s$).
>
> → It would also be interesting to see why taking actions based on the student network enables better defense.
>
> In the case of UQP during test time the actions taken by the Q network are successfully attacked with a high probability by FGSM.  Essentially they are fooled into taking any random action except the one which they think is correct.  The student network on the other hand was able to be defended against FGSM perturbations and more often take the same action that it would have taken had without the perturbation applied.

---

### Official Review · AnonReviewer3 · 2018-11-05
**Incremental Novelty Presentation Needs Major Improvements**

**Rating:** 3
**Confidence:** 2

**Review:**

Stating the observation that the RL agents with neural network policies are likely to be fooled by adversarial attacks the paper investigates a way to decrease this susceptibility.   Main assumption is that the environment is aware of the fact that the agent is using neural network policies and also has an access to those weights. The paper introduces a poisoning attack and a method to incorporate defense into an agent trained by DQN.  Main idea is to decouple the DQN Network into what they call a (Student) policy network and a Q network and use the policy network for exploration. This is the only novelty in the paper. The rest of the paper builds upon earlier ideas and incorporates different training techniques in order to include defense strategies to the DQN algorithm. This is summarized in Algorithm 1 called DadQN. Both proposed training methods; adversarial training and Provable robust training are well known techniques. The benefits of the proposed decoupling is evidenced by the experimental results. However, only three games from the Atari benchmark set is chosen, which impairs the quality of the evidence. In my opinion the work is very limited in originality with limited scope that it only applies to one type of RL algorithm combined with the very few set of experiments for supporting the claim fails to make the cut for publication.

Below are my suggestions for improving the paper.
1. Major improvement of the exposition
  a. Section 2.2 Agent Aware Game notation is very cumbersome. Please clean up and give an intuitive example to demonstrate.
  b. Section 3 title is Our Approach however mostly talks about the prior work. Either do a better compare contrast of the underlying method against the  previous work with clear distinction or move this entire discussion to related work section.
2. Needs more explanation how training with a defending strategy can achieve better training rewards as opposed to epsilon greedy.
3. Improve the exposition in Tables 1 and 2. It is hard to follow the explanations with the results in the table. User better titles and highlight the major results.
4. Discuss the relationship of adversarial training vs the Safe RL literature.
5. Provide discussions about how the technique can be extended into TRPO and A3C.

---

> ### Author Response · Authors · 2018-11-26
> **Responses**
>
> →  Main idea is to decouple the DQN Network into what they call a (Student) policy network and a Q network and use the policy network for exploration. This is the only novelty in the paper.
>
> We also introduce an efficient attack, the UQP able to disrupt training for reinforcement learning.  To date, this paper is the first which is able to explicitly defend a deep RL agent against arbitrary adversarial attacks, and introduces the first neural RL agent (which we are aware of) to be formally verified robust.
>
> → However, only three games from the Atari benchmark set is chosen, which impairs the quality of the evidence.
>
> Each datapoint in each table roughly corresponds to 20 hours of GPU time, making more games quite expensive.  Other papers on this subject often report similar numbers of games [Behzadan et al. (2017), Mandlekar et al. (2017), Huang et al. (2017) ].  Nevertheless, we have included one more game, Breakout, and plan on including more.
>
> → Major improvement of the exposition
>
> We have made the suggested changes
>
> → Needs more explanation how training with a defending strategy can achieve better training rewards as opposed to epsilon greedy.
>
> Noisy DQN is considered to be the state of the art for exploration with DQNs and typically achieves better or as good as results than epsilon greedy, as shown additionally in the RainbowDQN paper.
>
> → Provide discussions about how the technique can be extended into TRPO and A3C.
>
> While we do plan on extending the student style learning to actor-critic methods, this remains a challenge as they often rely on the on-policy nature of the agent to draw samples from a distribution that it prescribes. Provable defenses and distillation techniques prescribe that certain losses, often based the argmax of the teacher’s policy, be used to train the student.  When such a loss is used, the samples drawn from playing the game no longer match those required by methods such as A3C and PPO.   While applying our technique in this setting is not yet considered, we have updated the appendix of the paper to demonstrate that A3C and PPO are both susceptible to a form of the UQP attack applied to only one frame, and in fact often fail to learn altogether in its presence.

---

### Official Review · AnonReviewer4 · 2018-11-10
**Promising but minor algorithmic contribution**

**Rating:** 5
**Confidence:** 4

**Review:**

The goal of this paper is to train deep RL agents that perform well both in the presence and absence of adversarial attacks at training and test time. To achieve this, this paper proposes using policy distillation. The approach, Distilled Agent DQN (DaDQN), consists of: (1) a "teacher" neural network trained in the same way as DQN, and (2) a "student" network trained with supervised learning to match the teacher’s outputs. Adversarial defenses are only applied to the student network, so as to not impact the learning of Q-values by the teacher network. At test time, the student network is deployed.

This idea of separating the learning of Q-values from the incorporation of adversarial defenses is promising. One adversarial defense considered in the paper is adversarial training -- applying small FGSM perturbations to inputs before they are given to the network. In a sense, the proposed approach is the correct way of doing adversarial training in deep RL. Unlike in supervised learning, there is no ground truth for the correct action to take. But by treating the teacher's output (for an unperturbed input) as ground truth, the student network can more easily learn the correct Q-values for the corresponding perturbed input.

The experimental results support the claim that applying adversarial training to DaDQN leads to agents that perform well at test time, both in the presence and absence of adversarial attacks. Without this teacher-student separation, incorporating adversarial training severely impairs learning (Table 2, DQN Def column). This separation also enables training the student network with provably robust training.

However, I have a few significant concerns regarding this paper. The first is regarding the white-box poisoning attack that this paper proposes, called Untargeted Q-Poisoning (UQP). This is not a true poisoning attack, since it attacks not just at training time, but also at test time. Also, the choice of adding the *negative* of the FGSM perturbation during training time is not clearly justified. Why not just use FGSM perturbations? The reason given in the paper is that this reinforces the choice of the best action w.r.t. the learned Q-values, to give the illusion of successful training -- but why is this illusion important, and is this illusion actually observed during training time? What are the scores obtained at the end of training? Table 1 only reports test-time scores.

In addition, although most of the paper is written clearly, the experiment section is confusing. I have the following major questions:
- What is the attack Atk (Section 4.3) -- is it exactly the same as the defense Def, except the perturbations are now stored in the replay buffer? Are attack and defense perturbations applied at every timestep?
- In Section 4.2, when UQP is applied, is it attacking both at training and at test time? Given the definition of UQP (Section 2.4), the answer would be yes. If that’s the case, then the "none" row in Table 1 is misleading, since there actually is a test time attack.

The experiments could also be more thorough. For instance, is the adversarial training defense still effective when the FGSM \epsilon used in test time attacks is smaller or larger? Also, how important is it that the student network chooses actions during training time, rather than the teacher network? An ablation study would be helpful here.

Overall, although the algorithmic novelty is promising, it is relatively minor. Due to this, and the weaknesses mentioned above, I don't think this paper is ready for publication.

Minor comments / questions:
- Tables 1 and 2 should report 95% confidence intervals or the standard error.
- It’s strange to apply the attack to the entire 4-stack of consecutive frames used (i.e., the observations from the last four timesteps); it would make more sense if the attack only affected the current frame.
- For adversarial training, what probability p (Section 3.2) is used in the experiments?
- In Section 4.2, what does “weighted by number of frames” mean?
- In which experiments (if any) is NoisyNet used? Section 4.1 mentions it is disabled, and \epsilon-greedy exploration is used instead. But I assume it’s used somewhere, because it’s described when explaining the DaDQN approach (Section 3.1).

---

> ### Author Response · Authors · 2018-11-26
> **Responses Part 1**
>
> → The Untargeted Q-Poisoning (UQP) is not a true poisoning attack since it attacks not just at training time, but also at test time.
>
> While UQP is designed to be most effective when used at both training and test time, our evaluation shows that it is still quite effective when used only during training, thus making it functionally a poisoning attack.  In table 1, the DQN is given the UQP attack during training, and is given either no attack, or the UQP/FGSM attack during testing.  Even when given no attack during testing, it is apparent the DQN has failed to learn, and thus has been poisoned.  Figure 3 also demonstrates that learning itself is hindered when given the UQP attack.
>
> → The reason given in the paper is that this reinforces the choice of the best action w.r.t. the learned Q-values, why is the illusion of successful training important to the UQP attack, and is it actually observed during training time?
>
> The illusion mentioned would be an effect on loss (which is typically not an interpretable value), and not score.  Technically, the negative used here decreases the loss to the already chosen target value making an agent (or even just a classifier) see what it believes is already more typical, and thus essentially decreases the diversity of the test data.
>
> → What are the scores obtained at the end of training? Table 1 only reports test-time scores.
>
> While Table 1 only reports test-time scores, it does so for validation games without any attack.  Further, figure 3 is a graph of training-time scores over time which demonstrates the score is indeed reduced. The reported scores are scores from the best version of the net over an entire training run.
>
> → Is Atk the same as the defense Def, except the perturbations are now stored in the replay buffer?
>
> Yes.  This is explained in “Attacks & Defenses” in section 4.1.   We have highlighted this.
>
> → Are attack and defense perturbations applied at every timestep?
>
>   We have made it clearer that both Atk and Def use FGSM with p=0.4.
>
> → When UQP is applied, is it attacking both at training and at test time?
>
> UQP is defined to be aware of whether the agent is learning or testing, the definition does not specify that it must always be used. Our experiments examine the case where one attack is used to train and another (in some cases the same or a functionally equivalent attack) is used during testing. During testing, UQP and FGSM have precisely the same behavior, but during training they have different behavior.  As such, it really only makes sense to talk about UQP when training.  To clarify, we have pointed this out in the experimental section of the paper.
>
>
> → Is the adversarial training defense still effective when the FGSM epsilon used in test time attacks is smaller or larger?
>
> More often than not, adversarial defenses train with the same epsilon attacks with which they are evaluated against [Wong & Kolter (2018), Mirman et al. (2018), Madry et al. (2018)].  As reinforcement learning is particularly computationally expensive, we believe classification tasks would be better suited to exploring training and attack asymmetry in detail.
>
> → Also, how important is it that the student network chooses actions during training time, rather than the teacher network?
>
> This can be seen in both Table 1 and Table 2 by comparing the results for the DQN with a training attack column with the results for DadQN with a training attack and a defense column.  The results there are either similar, or significantly better when using the student to pick actions, even when no attack is used at test time or a similar attack is used at test time as during training.  As the validation games here are either similar to the training games or easier, one can observe that learning is hampered in the DQN when you do not use a student to pick actions.
>
> → Tables 1 and 2 should report 95% confidence intervals or the standard error.
>
> As stated in the text, Table 1 shows the average final episode score (weighted by number of frames) from the 15 consecutive best (by sum score) validation games during training. This is the maximum of a weighted aggregation over a series of data. Suitable uncertainty measures would require to repeat the experiment and then give statistics (mean + confidence interval or standard deviation) over the repetitions. However, as one experiment might take up to 30 GPU hours we were not able to perform repetitions and gather an uncertainty measure for the numbers.

---

> ### Author Response · Authors · 2018-11-26
> **Responses Part 2**
>
> → It’s strange to apply the attack to the entire 4-stack of consecutive frames used.
>
> Single frame attacks conceptually fit the setting, but prior work [Behzadan et al. (2017a)] uses 4-stack attacks. Other works [Huang et al. (2017),  Lin et al. (2017b), Behzadan et al. (2017b)] do not specifically highlight whether they use single or 4-stack approaches, but from their descriptions it appears that they attack 4-stacks. This impression is given as they describe the usage of standard attacks such as FGSM or Carlini & Wagner’s [6] attacks applied to the DQN. The standard forms of the attacks are always performed with respect to the full input of the network, which in the case of DQN are 4-stacks.
>
> In our setting, as we are primarily evaluating the effectivity of defense, we chose to use the attack which was most powerful.  We have updated the paper’s Appendix to include results from applying the UQP attack to A2C and PPO on only a single frame.  Even though the attack is only applied to one frame here, the networks are even more affected and fail to learn at all.
>
> → For adversarial training, what probability p is used in the experiments?
>
> We use a probability of defense of 0.4 for any FGSM defense.  We have updated the paper to say this in section 4.1 “attacks & defenses.”
>
> → What does “weighted by number of frames” mean?
>
> An equivalent way of saying this would be that the numbers are the average per-frame score from 15 full episodes.  Some games, such as roadrunner, have multiple levels which is a non trivial problem for DQNs. As such, when the agent’s achieved a certain score, it then proceeded to play episodes it wasn’t as adept at. While these episodes achieved a low score, it would be unfair to compare directly against an agent which had in fact never reached the second level. Thus we introduced the weighing.
> In the updated paper’s Appendix B, we also include the unweighted average per game which demonstrates that the defense is still effective as well as a mathematical discussion of the weighting.
>
> → In which experiments is NoisyNet used? Section 4.1 mentions it is disabled
>
> All. “During these episodes we disable noise due to NoisyNet and use epsilon-greedy exploration with  = 0.005.” has a typo.  It should be “validation episodes” and not “these episodes”  This has been fixed in the revision.

---

### Author Response · Authors · 2018-11-26
**Reply to all**

We thank the reviewers for their comments. To address the reviews we extended the paper with various clarifications and improvements as well as new experiments and more data metrics from the existing experiments:

- Added an appendix which shows that the UQP attack prevents actor-critic algorithms (A3C and PPO) from training.

- The related work section has been promoted to earlier in the paper and has been significantly extended to discuss more of the non-RL robustness literature as well as other works covering robustness in the RL settings.

- In the appendix we explain and justify the use of frame-weighted scores in the evaluation and give the unweighted data for the results of table 1.

After thorough review of related work and implementing new experiments, to our knowledge DadQN remains the only algorithm to defend against the UQP attack. While prior work does discuss adversarial training, our work is the first to modify the learning algorithm itself for the purpose of defense rather than augment the training set.  In particular, prior work in safe reinforcement learning for deep networks has either:

- Proposed adversarially perturbing the input to DQNs, e.g., Behzadan & Arslan (2017) proposed standard adversarial training against FGSM-style attacks.  While their method can be effective, it often hinders learning. We show that these are not sufficient defenses for DQNs, but can be used to defend the  DadQN and while only minimally impacting performance.

- Proposed adversarially perturbing the input to deep policy learning methods when the parameter space is small. In particular prior work has focused on control tasks (Pinto et al. (2017)), physically plausible perturbations (Mandlekar et al. (2017)) or adversarial training to improve algorithm performance (Gu et al. (2018)). -- All of which are not studied in agent-aware attacks such as FGSM or deal with much smaller parameter spaces where these attacks are less efficient.

- Explored whether an already learned policy could be attacked.   Huang et al. (2017), Lin et al (2017) and Behzadan et al. (2017) have collectively demonstrated the vulnerability of multiple deep RL algorithms to a variety of attacks.

We hope that this improves the readability of the paper and offers better understanding of the proposed algorithm. Further we have replied to each reviewer individually and will gladly answer further questions.

---

### Meta-Review · Area_Chair1 · 2018-12-15
**limited novelty**

**Confidence:** 5
**Recommendation:** Reject

**Metareview:**

Reviewers had several concerns about the paper, primary among them being limited novelty of the approach. The reviewers have offered suggestions for improving the work which we encourage the authors to read and consider.